# Insights on 21 Years of HBV Surveillance in Blood Donors in France

**DOI:** 10.3390/v14112507

**Published:** 2022-11-12

**Authors:** Pierre Cappy, Laure Boizeau, Daniel Candotti, Sophie Le Cam, Christophe Martinaud, Josiane Pillonel, Martin Tribout, Claude Maugard, Josiane Relave, Pascale Richard, Pascal Morel, Syria Laperche

**Affiliations:** 1Department of Microbiology, Henri Mondor University Hospital, Assistance Publique-Hôpitaux de Paris, 94010 Créteil, France; 2National Reference Centre for Transfusion Infectious Risks, Institut National de la Transfusion Sanguine (INTS), 75015 Paris, France; 3Etablissement Français du Sang (EFS), 93210 La Plaine St-Denis, France; 4Centre de Transfusion des Armées (CTSA), 92140 Clamart, France; 5Santé Publique France, 94410 Saint Mau, France; 6Etablissement Français du Sang Occitanie (EFS Occitanie), 31059 Toulouse, France; 7UMR 1098 RIGHT, INSERM, Université de Franche-Comté, 25000 Besançon, France

**Keywords:** HBV, blood donations, NAT, occult B infection, acute HBV infection

## Abstract

Hepatitis B virus (HBV) infection is the most frequent viral infection found in blood donors (BDs) in France. We analyzed the epidemiological and sero-molecular data on HBV infection gathered over the past two decades by the French haemovigilance surveillance network, blood screening laboratories, and the national reference center for transfusion infectious risks (NRC). Between 2000 and 2020, 6149 of the 58,160,984 donations (1.06/10,000) tested HBV positive, 98% of them from first-time blood donors (FTBDs). In addition, 2212 (0.0071%) of the 30,977,753 donations screened for HBV DNA tested DNA positive, of which 25 (1.1%) were positive only for this marker. HBV prevalence decreased by 2.8-fold and the residual risk for transfusion-transmitted HBV infection decreased 13-fold and was divided by 13. The major risk factor for HBV infection was the origin of donors (endemic country, 66.5%), followed by parenteral exposure (10.7%). In the whole HBV-positive BD population, genotype D was predominant (41.8%), followed by genotypes A (26.2%) and E (20.4%), reflecting the geographical origin of donors. The low and decreasing prevalence and incidence of HBV infection in French BDs, coupled with a screening strategy using three HBV markers (HBsAg, anti-HBc and DNA), ensures a high level of blood safety, further reinforced by the implementation of pathogen-reduction measures.

## 1. Introduction

Hepatitis B virus (HBV) infection is a major global public health issue, with more than two billion people worldwide who have been in contact with the virus, and over 296 million of these living with a chronic infection [1]. Although antiviral agents have been shown to effectively suppress viral replication, HBV cure remains a challenge because of the persistence of covalently closed circular DNA (cccDNA), HBV-DNA integration into the host genome, and impaired immune response [2]. Vaccination is a key intervention for durably preventing HBV infections [3,4], even though some vaccination breakthrough infections have been reported [5]. In high endemic areas such as Sub-Saharan Africa, South East Asia, China, and South America, up to 50% of people have been infected and over 8% are chronic carriers, defined by the persistence of HBV surface antigen (HBsAg) for more than 6 months [1]. In France, despite being classified as a low endemic country with HBsAg prevalence below 2%, HBV remains the most frequently detected virus in blood donors, in association with the ethnical backgrounds of infected donors. In an extensive study carried out over a 9-year period (1999–2007), we previously showed that HBV infected blood donors mainly originated from endemic countries, especially Sub-Saharan Africa [6]. Accordingly, the circulation of HBV genotype E, originated from this geographical area, was detected in 17.5% of blood donors in France between 2005 and 2007.

Several measures have been introduced over time to reduce the risk of HBV transmission by transfusion. The pre-donation evaluation consisting of a self-administrated written questionnaire, including a range of risk factors for transfusion-transmitted infections (TTI) and an interview with medical staff, is the first line to prevent the risk of infected blood entering the blood supply. However, as the main risk factor for HBV infection in donors was being from or related to an endemic country, the pre-donation selection is poorly efficient for HBV. Blood screening is the second-line to mitigate the risk. In France, the HBV blood screening strategy has included testing for HBsAg since 1971, antibodies to the HBV core protein (anti-HBc) since 1988, and nucleic acid testing (NAT) in individual donation since 2005 in the army blood service, 2006 in overseas territories, and since 2010 in the entire country.

HBsAg remains the key marker for HBV screening in blood donors. Several assays have been developed over time, and enzyme immunoassays (EIAs) are the most commonly used assays for blood screening. Even though HBV mutations associated with HBsAg structural changes, and circulating immune complexes between HBsAg and hepatitis B surface antibodies (anti-HBs) may negatively affect the performance of HBsAg detection [7,8], these assays globally show a high analytical sensitivity level. The cost and equipment requirements of these assays may, however, limit their use in resource-limited settings. To overcome this shortcoming, several reliable and sensitive rapid diagnostic tests have been successfully developed [9]. They remain less sensitive than EIAs but are easy to perform, and do not require expensive equipment and facilities. These assays are also used as pre-donation screening in some high endemic countries to avoid costly blood collection, complementary testing, and finally the destruction of blood components [10,11].

Anti-HBc is a reliable marker of HBV natural exposure as it remains detectable throughout the entire course of infection (at least in immunocompetent individuals) and persists for life after recovery, even when all other HBV markers become undetectable. However, anti-HBc assays lack specificity and the absence of specific confirmatory tests makes it difficult to discriminate between true and false-positive samples [12]. Despite these limitations, anti-HBc screening has been implemented in HBV low-endemic countries where donor deferral was considered sustainable in terms of donation wastage. Conversely, anti-HBc testing cannot be implemented in high-endemic countries without compromising blood availability.

HBV-DNA detection in blood donations has been implemented in many countries worldwide to reduce the risk associated with infected blood donors without detectable serological markers [13]. When affordable, NAT assays with extensively documented sensitivity are used in blood banks, showing 95% limit of detection (LoD) below 10 IU/mL [14]. Different strategies are used worldwide: testing in minipool (MP) of six to 96 donations (MP-NAT) or in individual donation (ID-NAT) [14]. The implementation of NAT prevents transfusions from donors recently infected and in the pre-seroconversion “window period” (WP) defined as the time between infection and detection of viral antigen or antibodies, and from HBsAg-non reactive donors with persistent low-level chronic infection so-called occult HBV infection (OBI).

OBI is a form of chronic HBV infection where replication competent cccDNA persists in hepatocytes. This entity is defined as intermittently detectable low levels of HBV DNA (generally <200 IU/mL) in plasma with undetectable HBsAg, and with/without anti-HBc or anti-HBs [15]. OBI has been reported among healthy asymptomatic blood donors, patients with chronic liver disease, and patients with hepatocellular carcinoma [16]. The prevalence of OBIs tends to be higher in regions with high HBV prevalence and varies according to clinical situations [17]. A recent review reported an overall OBI prevalence at 0.82% (95% CI: 0.69–0.96) in the general population with variations from 4.25% (95% CI: 1.64–7.87) in haemodialysis patients and 5.14% (95% CI: 2.26–9.01) patients with other risk factors, up to 13.99% (95% CI:8.33–20.79) in patients with other liver diseases and 16.26% (95% CI: 10.97–22.34) in HIV patients [18]. Transfusion transmission of HBV with blood components from OBI donors has been reported with rates depending on viral loads and transfused plasma volumes, the presence of anti-HBs in donor and/or recipients, and the general immune status of the recipient [19,20,21,22,23,24,25,26,27,28,29]. The minimal infectious dose from OBI has been estimated to be 16 copies (or three IU) of HBV DNA [19]. OBI transmission can be prevented by a systematic anti-HBc screening, when possible, and/or HBV DNA NAT with high sensitivity level.

Finally, pathogen inactivation methods showed the capacity to reduce transfusion-transmitted infection (TTI). Different pathogen inactivation systems are currently applied to plasma and platelet concentrates [30], with an estimated inactivation yield of 2 to 4 log for HBV but none are yet available for red cell components, limiting the reduction of the risk to platelet and plasma units [31,32,33].

In the current study, based on the national surveillance of infected blood donors in France, we analyzed trends in epidemiology and virology of HBV positive (HBsAg+ and/or HBV DNA+) blood donors identified between 2000 and 2020 with the aim of providing insights on HBV infection in the general population. We also assessed the residual risk and reported some particular cases identified through our haemovigilance system.

## 2. Material and Methods

### 2.1. Study Population

The surveillance of blood donor population is performed at the national level (including overseas territories) with the objective to collect data from all donors who donated at the National Blood Service (Etablissement Français du Sang-EFS) and the army blood service (Centre de Transfusion des Armées-CTSA). All data from first-time (FTD) and repeat donors (RD) were retrieved from the National Epidemiological Donor database, which contains demographic characteristics (age, sex), probable source of infection, geographical origin, collected during post-donation interview, as well as virological characteristics of HBV positive donors, all together with the total number of donations per year. All HBV positive donation samples, when available, were sent to the National Reference Center for transfusion infectious risks (NRC) for further virological investigations. We analyzed the general epidemiological data gathered between 2000 and 2020, and the associated molecular data available after 2005.

### 2.2. HBV Donation Testing

All donations were individually tested for HBsAg (Prism or Architect Abbott Diagnostics; sensitivity 0.08–0.1 ng/mL), anti-HBc (Prism or Architect Abbott Diagnostics). HBV-DNA testing (nucleic acid testing, or NAT) was introduced progressively from 2005 using Procleix Ultrio HIV-1/HCV/HBV Assay on Tigris Platform (Grifols, Barcelona, Spain; 95% LoD95 of 10.9 IU/mL (7.5–17.8) [34] and Procleix Discriminatory HBV assay (LoD95 similar to the multiplex assay) to identify HBV-DNA in Procleix Ultrio positive samples. HBsAg and/or HBV-DNA positive donations irrespective of their anti-HBc status were subjected to further investigations at the NRC including anti-HBs (Murex anti-HBs assay, DiaSorin, Saluggia, Italy), viral load determination (Cobas Taq Man, Roche Diagnostic, limit of quantification: 6 UI/mL) and genotyping in the polymerase gene (*pol)* (sensitivity around 150 IU/mL) with a nested PCR using primers 5′ CCCTGCTCGTGTTACAGGCGG 3′ + 5′ GTTGCGTCAGCAAACACTTGGCA 3′ (external PCR) and 5′ GACTCGTGGTGGACTTCTCTCA 3′ + 5′ TGCCGRGCAACGGGGTAAAGG 3′ (nested PCR, size of the amplicon: 910 bp), and sequenced using primer 5′-TGGCCAAAATTCGCAGTCC (sense) and 5′-AAGGCCTTGTAAGTTGGCGA (anti-sense) or 5′ ACTTCTCTCAATTTTCTAGG 3 (sense) and 5′-CCCARAAGACCCACAAT (anti-sense) when the first set of primers yielded no sequence. If the amplification of pol was negative, a shorter sequence was amplified in *pol/HBsAg* (sensitivity lower than 30 IU/mL) using primers 5′ TCGTGGTGGACTTCTCTC 3′ + 5′ ACA GTG GGG GAA AGC CC 3′ (external PCR) and 5′ ACTTCTCTCAATTTTCTAGG 3′ + 5′ CTA CGA ACC ACT GAA CAA AT 3′ (nested PCR; size of the amplicon: 444 bp) and sequenced using primers 5′ TGGCCAAAATTCGCAGTCC 3′ (sense) and 5′ CTA CGA ACC ACT GAA CAA AT 3′ (anti-sense) or 5′-TGG CCA AAA TTC GCA GTCC 3′ (sense) and the same anti-sense primer, when the first set of primers yielded no sequence. Sequences were aligned with a set of HBV sequences from the 10 current genotypes (A to J) using MEGA (version used: MEGA 7) [35] and HBV genotypes were determined using phylogeny (neighbor-joining Kimura 2-parameter method, and 1000 bootstrap replicates to assess the reliability of the branching order) and the geno2pheno website [36].

Alternatively, when viral load was low, HBV DNA was purified from 10 to 20 mL of donor’s plasma after viral particle concentration by high-speed centrifugation [37].

### 2.3. Definitions

Acute HBV infection (ABI) was defined as the presence of HBV-DNA with or without HBsAg (the latter situation corresponding to NAT yield cases) in the absence of anti-HBc.

Chronic HBV infection (CHB) was defined as the presence of HBV-DNA, HBsAg and antibodies to anti-HBc [38]. Samples positive for HBsAg and anti-HBc but with undetectable HBV-DNA were also classified as chronic infections. Samples positive for HBV-DNA and anti-HBc without detectable HBsAg were classified as OBI. When HBV-DNA and anti-HBs were the only positive markers, samples were classified as anti-HBs only OBI, in the absence of seroconversion for HBsAg, anti-HBc, in follow-up samples and/or if HBV DNA and anti-HBs was detected in previous donations when retrospectively investigated.

Vaccine breakthrough in vaccinated donors or abortive infection were defined as the presence of detectable HBV-DNA and anti-HBs in the index donation, followed by a significant rise in anti-HBs titer and/or anti-HBc seroconversion in follow-up samples. If no anti-HBc seroconversion was evidenced or if no follow-up samples were available, samples with low anti-HBs titers (<150 mIU/mL) combined with detectable HBV DNA in donors with a history of HBV vaccination were classified in this category.

### 2.4. Residual Risk Estimates

HBV residual risk (RR) is the estimated risk of collecting a potentially infectious unit during the window period in which current tests cannot detect the presence of the pathogen. We used the window-period model for estimating the HBV residual risk [39,40], which is defined by RR = (observed incidence of HBsAg and/or HBV DNA in repeat blood donors calculated in 3-year periods (expressed as 100,000 person-years)) × marker-negative window period (22 days)) [41]. Seven periods of three years from 2000–2002 to 2018–2020 were considered. Repeat donors with a donation collected in a previous 3-year period who were found HBV-DNA positive in the lookback investigation were excluded from the calculation of the risk, as they were therefore not incident cases.

### 2.5. Lookback Investigations

According to our haemovigilance organization, in the case of seroconversion of repeat blood donors newly identified positive for HBsAg or HBV-DNA, lookback investigations were undertaken in previous donations when available. This included retrospective testing (HBV-DNA with Cobas Taq Man, Roche Diagnostic, Meylan, France) of archived samples (preserved during 3 years after donation), tracing components, and recipients’ investigations when appropriate. If the previous donation tested positive for HBV DNA, subsequent investigations were performed on all previously available archived samples.

### 2.6. Statistical Analysis

Population characteristics were reported as absolute numbers with percentages, and medians with minimum and maximum values, as appropriate. Statistical analyses were computed using the GraphPad Prism 8 package. Differences between median were tested using the Mann-Whitney test, and between distributions regarding demographic data using a χ2 test for distribution, with Yates’ correction whenever calculated frequencies were <5%, or Fisher exact test to compare only two categories. Results were significant whenever *p*-value was <0.05.

## 3. Results

### 3.1. HBV Epidemiology in Blood Donors in France (2000–2020)

Between 2000 and 2020, 6149 of the 58,160,984 donations (1.06/10,000) tested HBV reactive (HBsAg+ and/or HBV DNA+). The rate of HBV positive donations decreased 4.8-fold (1.77/10,000 in 2000 vs. 0.37/10,000 donations in 2020) in the whole-blood donor population, and 3.5-fold (10.3/10,000 in 2000 vs. 2.9/10,000 donations in 2020) in first-time blood donors (FTBD) (Figure 1). This rate remained stable around 0.02/10,000 donations in the repeat blood donor (RBD) population. Of the 6149 HBV positive donations, 6025 (98%) were from FTBDs for an overall prevalence of 8.13/10,000 donors. The prevalence decreased 2.8-fold between 2000 and 2020 (Figure 1, Appendix A). The incidence of HBV infection, estimated over three consecutive 3-year periods among repeat blood donors who seroconverted for HBsAg and/or HBV DNA ranged from 1.66/100,000 person-year (PY) in 2000–2002 to 0.26/100,000 PY in 2018–2020 (Appendix A).

Over the whole study period, 4146 (68.8%) of the 6025 FTBD returned to the blood center to be interviewed in order to investigate the potential infection source. Among them, 10% reported no risk factors, 66.5% originated from a high endemic area (mainly Sub-Saharan Africa), 10.7% had possible parenteral exposure (nosocomial, surgery, tattoo, etc., …), 6.2% reported familial exposure, 4.3% had probable sexual exposure, and 2.3% had stayed or travelled in a high endemic area. Although subject to some fluctuations over time, being born in an endemic area remained the main risk factor among HBV-infected FTBDs in France, suggesting the acquisition of the infection early in life and first diagnosis at the time of donation. Ninety-eight (79.7%) of the 123 RBDs returned to the blood center for a post-donation interview: 30.7% of them reported no risk factors while 30.7% reported a sexual risk, 17.2% a parenteral exposure, 11.2% originated from a endemic country, 6.1% had stayed in a endemic country and 4.1% reported a familial exposure.

### 3.2. HBV Molecular Epidemiology (2005–2020)

As shown in Figure 2a, the sequencing of 3267 strains (82.5%) from the 3959 HBV positive donations collected between 2005 and 2020 showed that donors were predominantly infected with genotype D (41.8%), followed by genotype A (26.2%), and genotype E (20.4%). A significantly lower prevalence of the other genotypes was observed: 5.7% of genotype B, 5.1% of genotype C, and 0.7% of genotype F. This distribution was stable over time. When the geographical origin of the donor was reported (*n* = 3064), HBV genotype and subtype distribution reflected donor ethnicity (Figure 2b–d): subtypes A1, A3, and A4 and genotype E were predominantly found in donors from Sub-Saharan Africa, genotype D (especially D1 and D7–8) in those of Mediterranean basin, and genotypes B and C from Asia.

### 3.3. Characterization of Donations according to HBV Markers (2005–2020)

After NAT implementation, 30,977,753 donations were screened for HBV DNA between 2005 and 2020, of which 2212 (0.0071%) were reactive for HBV DNA and were broken down as follows: 2079 (94.1%) HBV DNA+/HBsAg+/anti-HBc+ (CHB), 81 (3.6%) HBV DNA+/HBsAg-/anti-HBc+ (OBI), 27 (1.2%) HBV DNA+/HBsAg+/anti-Hbc− (ABI), and 25 (1.1%) HBV DNA+/HBsAg-/anti-Hbc− (acute infection or seronegative OBI) (Table 1). These proportions remained stable over time during the study period. Of note, the NAT was introduced in metropolitan France only in 2010, compared to the overseas territories and the army blood donors in 2005 and 2006, respectively, hence the differences with the data from the global genotyping data mentioned above.

### 3.4. Comprehensive Analysis of Donations Exhibiting Discrepant Molecular and Serological HBV Markers

Of the 81 donors with anti-HBc positive OBIs (HBV DNA+/HBsAg−/anti-HBc+), 69 (85%) were FTBDs and 11 (15%) were RBDs (one had an unknown status), the sex ratio was 5.2 (68 males/13 females), median age was at 47 years (20–69), median VL was <0.70 log IU/mL (range: <0.70–3.03 log IU/mL). Among the 48 donors tested for anti-HBs, 24 (50%) were positive with low titers (median: 55 mIU/mL, range: 11–311 mIU/mL) (Table 1). Risk factors for HBV infection were identified in 60 (74%) donors. The most common risk factor was place of birth or travel to a highly endemic area (*n* = 41 (68%)), followed by parenteral or nosocomial exposure (*n* = 7 (12%)), sexual exposure (*n* = 7 (12%)) or other or unknown exposure (*n* = 5 (8%)). HBV sequences could be obtained for 44 OBI strains (missing data due to insufficient sample volume or low viral load) and showed a genotype distribution similar to those observed in the whole HBV-infected donor population with genotype D being dominant (*n* = 21 (48%)), then genotypes A (*n*= 10 (23%)), E (*n*= 9 (20%)), B (*n*= 2 (5%)), C (*n*= 1 (2%)), and F (*n*= 1 (2%)).

Among the 27 donors with probable acute infection (HBV DNA+/HBsAg+/anti-Hbc−), 18 (63%) were RBDs, 7 (26%) FTBDs (two with unknown data). The sex ratio was 4.0 (20 males/5 females, 2 unknown data), the median age was at 47 years (19–68 years), the mean VL was 5.24 log IU/mL (range 2.24–8.04 log IU/mL) (Table 1). Of the 21 donors interviewed post-donation, eight (38%) reported no risk factor, whereas a probable mode of infection was identified for 13 (62%) donors, eight of whom reported sexual risk and five parenteral exposure. The most prevalent genotype was genotype A (*n* = 13 (55%); three subtypes A1, seven subtypes A2, 3 not subtyped), then genotypes E (*n* = 4 (18%)), B (*n* = 2 (9%); 1 subtype B1), D (*n* = 2 (9%]) one subtype D1, one subtype D7), C1 (*n* = 1 (5%)) and F1 (*n* = 1 (5%)).

Twenty-five donors presenting an HBV DNA+/HBsAg−/anti-Hbc− profile were further analyzed (Table 2). Among these donors, 15 (60%) were RBDs and 10 (40%) FTBD, sex ratio was 1.3 (14 males/11 females), and median age 28 years (18–68 years). The median HBV DNA load available for 18 samples was 1.04 log IU/mL (range: <6–456 IU/mL). A possible risk factor could be identified for 14 donors: eight had an HBV-positive partner or originated from a highly endemic country, four reported sexual exposure, one reported piercings, and one lived in an endemic country during childhood. These HBV DNA+/HBsAg-/anti-Hbc− could be tentatively associated with a different stage of HBV natural infection based on additional anti-HBs testing at index time and serological follow-up. No data were available for four samples (Table 2, donors 3, 9, 12, and 18) and six anti-HBs-negative samples (Table 2, donors 1, 5, 7, 14, 19, and 24) at index time, including four RBDs and two FTBDs, who were identified as probable acute infection defined by HBsAg and/or anti-HBc seroconversion in follow-up. The mean age was 34 years (19–58 years). Interestingly, the median VL was 28 IU/mL (range: <6–456 IU/mL) that significantly differed from the VL observed in HBV DNA+/HBsAg+/anti-Hbc− donors as indicated above. A risk of HBV sexual exposure or partner originating from an endemic area was reported for three donors. HBV genotyping was performed in five samples and identified two strains genotype A, two E, and one F.

Complementary anti-HBs testing identified 12 reactive samples (median titre: 120 IU/L; range: 11–414 IU/L) that were considered as isolated anti-HBs OBI at index time (samples ID: 2, 6, 8, 11, 13, 15, 16, 17, 20, 22, 23, and 25). The mean age was 33 years (18–68 years) and the median VL was 11 IU/mL (range: 4–234 IU/mL). Seven (58.3%) of these donors reported HBV vaccination. In addition, three samples (4, 10, 18) were anti-HBs non-reactive despite two of them who reported vaccination, and could be considered as seronegative OBI. Data were missing for four samples at index time (ID: 3, 9, 12, and 21). Furthermore, anti-HBc seroconversion in five HBV DNA+/HBsAg-/anti-Hbc− donors (ID: 4, 8, 13, 16, and 20) and a substantial increase of anti-HBs levels (188–8000 IU/L) without HBsAg or anti-HBc seroconversion in three donors (ID: 10, 11, and 21) at follow-up suggested probable recent HBV vaccine breakthrough infection, likely abortive in the latter three cases. This was supported further by vaccination being definitively documented in five cases. Vaccine breakthrough infection could be also suspected in three other vaccinated donors (ID: 15, 17, and 23) who did not seroconvert to anti-HBc or HBsAg, although no anti-HBs data were available at follow-up (13 days to 8 months). A similar profile was observed for a fourth unvaccinated donor (ID: 22). No follow-up data were available for three donors (ID: 2, 6, and 25), one of whom was vaccinated.

The acute HBV infection population was statistically different from the OBI population regarding donor status, with more likely repeat donors (*p* < 0.0001) (HBV infection acquired between two donations), higher viral loads (*p* < 0.0001), parameters compatible with a primary HBV infection.

The HBV DNA+/HBsAg−/anti-Hbc− population was statistically different from the OBI population regarding donor status, with more likely repeat donors (*p* < 0.0001), donor age, with younger donors (*p* < 0.0001) (HBV infection acquired between two donations), and higher viral loads (*p* < 0.0021). When compared to the acute infections, only the viral load was significantly different, very low viral load in the DNA+/HBsAg−/anti-Hbc− population (*p* < 0.0001), and a clear trend for younger age was observed for the latter population (*p* = 0.0616).

### 3.5. Seronegative OBI: A Case Report

One unvaccinated donor (Table 2, donor 22) who tested HBV DNA+/HBsAg-/anti-Hbc at index and follow-up (39 days) was considered as anti-HBc seronegative OBI.

The donor was a RBD, who donated in July 2019, and found non repeat-reactive (NRR) with NAT (1+/4−), without HIV, HBV or HCV associated serological markers. He had given 53 times (whole blood (*n* = 29) and plasma (*n* = 24)) before this positive index donation (D). Non-repeat reactive results were observed in two subsequent follow-up samples (C1, C2) with a positive NAT discriminatory HBV assay. HBV DNA detection, performed on D, C1, C2 and on nine previous donations (N-1 to N-9) collected between 2016 and 2018, was positive in seven samples (N-8, N-5, N-4, N-3, N-1, C1 and C2) with unquantifiable VL (<6 IU/mL). S region was genotyped in 5/7 samples (A2 subtype) and a full-length sequence was obtained after high-speed centrifugation of 10 mL of C2 plasma. No specific mutation was found in preC/C, S, POL or X, compared to a non-OBI genotype A2 consensus sequence derived from the alignment of 41 HBV genotype A2 complete sequences previously obtained from HBsAg+/NAT+ blood donors [42]. HBsAg could be quantified around 6 IU/L in D using ultra-sensitive HBsAg assay (Lumipulse HBsAg). Although anti-HBs were found around 600 IU/L in C1 and C2, confirming the serology results obtained in 2009 (275 IU/L), the donor denied being vaccinated against HBV. Lookback studies performed on the nine previous donations showed that 17 blood products had been transfused: seven red blood cell units, six platelet units and four plasma units. Eight recipients could not be reached, six died before the study, two were anti-HBs+, and one was HBV negative.

### 3.6. HBV Residual Risk

Based on the incidence/WP model, the residual risk was 1/9,400,000 donations (0–1/2,300,000) for the 2018–2020 period. This risk decreased 13-fold when compared to the 2000–2002 period (1/700,000 donations (1/1 220,000–1/150,000)) NAT testing being not yet implemented and WP length being 38 days at that time (Figure 3).

## 4. Discussion

This study aims to characterize HBV infection in 6149 blood donations tested HBV positive for HBV (HBsAg+ and/or HBV-DNA+) in France over a 21-year period. Although blood donors are selected at the pre-donation stage, they represent a sentinel population that provides valuable information on infections in the general population, especially when this latter is not systematically surveyed. In France, only acute HBV infections are subjected to a specific mandatory surveillance in the general population which nevertheless suffers from a lack of comprehensiveness (estimated to be 27% in 2016) [43,44]. The surveillance of blood donors, which has been in place for many years, allows for highlighting trends in epidemiological indicators and virological characteristics in an apparently healthy population. The data reported in this study indicates declining prevalence of HBV infection that was 2.8 times lower in 2020 than in 2000 (3.9/10 000 vs. 11.2/10 000 donors). Notably, in 2016, the prevalence in blood donors was five times lower than those estimated in the general population in a study based on home blood self-sampling and screening history (5.8/10,000 vs. 30/10,000) 40, possibly reflecting the impact of pre-donation selection.

The main risk factor identified at post-donation medical interview was the origin of the donor (endemic areas, mainly Sub-Saharan Africa). Similar data were also recently reported in the UK where the prevalence in blood donors of Asian and Sub-Saharan African ethnicity was significantly higher than in donors of other ethnic groups (eight to 22 times for Asian donors and 24 to 52 times for Black donors according to years) [38]. Unfortunately, we were not able to estimate the prevalence by ethnicity because these data were not recorded in whole-blood donor population. Although being born or having a relative from an endemic area has been reported as the most predictive factors for HBV infection in chronically infected blood donors, recently acquired infections were most frequently associated with a sexual risk behavior [45,46,47], as observed in our study in more than 50% of donors with acute infections.

In line with the knowledge of the genotypic distribution of HBV observed in chronic carrier in Europe [48], the genotype D was the most common genotype detected in our study population, followed by genotypes A and E.

Conversely, acute infections were associated with sub-genotype A2 as observed in other Western European countries [47,49,50] and in the USA [51], but not in Mediterranean basin where the genotype D remained predominant [52].

HBV DNA detection was introduced in blood donations to reduce the risk of viral transmission via transfusion. Between 2005 and 2020, we identified 25 donors among approximately 31 million blood donations tested who were positive for HBV DNA and negative for all other marker corresponding to 0.81 HBV NAT yield cases per million blood donations. In countries where donations were also tested for HBsAg, anti-HBc and HBV-DNA, the HBV-NAT yields were of the same order of magnitude and varied from 0.64 per million in Germany (46 million tested donations from 2008 to 2015) [14], to 0.97 per million in the USA (33 million donations tested from 2009 to 2015) [53]. The NAT yield is, however, higher in countries that did not implement anti-HBc testing to maintain the blood supply in a context of high endemicity, as HBV DNA positive but HBsAg negative also included anti-HBc positive OBIs.

Our estimates showed that the transfusion HBV residual risk is currently very low (1/9.4 million donations i.e., 1/3.5 years) due to the introduction of several measures that prevent infected donations from entering the blood supply. Blood screening based on three different HBV markers (HBsAg, anti-HBc and NAT), as performed in France, allows the detection of the large majority of HBV infection stages. The use of very sensitive screening assays is key to blood safety. Notably, viral loads of the 18/25 HBV-DNA only positive donations were very low (mean 58 UI/mL) justifying the use of a very highly sensitive NAT assay in individual testing to be able to detect early acute infection as well as OBIs. A study performed in South Africa showed that a NAT assay 2.4-fold more sensitive than a comparative assay detected twice more window-period NAT yield donations and 1.7-fold more occult HBV infections [54]. With the use of the highest sensitive assay, the reduction of the infectious WP was estimated at 10 days [54].

When HBsAg and anti-HBc serology in combination with ID-NAT are used, the residual risk would remain essentially associated with the DNA-negative eclipse phase in early acute infection and the extremely rare cases of isolated anti-HBs OBI, although no transmission was reported from such cases so far. Of the 25 NAT yield cases observed between 2007 and 2020, two were considered as anti-HBs only OBI (donors 22 and 23, Table 2) with VLs below 6 IU/mL. Donor 22 had no record of vaccination and was retrospectively detected HBV DNA positive at extremely low level in five of the nine archived samples collected over the 3 years prior to the index donation, supporting that HBV DNA detection can prove particularly challenging for OBI donors with very low HBV DNA levels that may be only intermittently detectable over time even by ID-NAT [55]. Notably, the lookback investigations did not evidence transfusion transmission via blood components from these blood donations, at least in recipients who could be tested. In contrast, donor 23 reported vaccination during childhood in China. Unfortunately, follow-up of donor 23 was limited to 13 days and the possibility of acute breakthrough infection could not be totally ruled out. However, infection with an HBV strain of Asian genotype B strongly suggested breakthrough infection in childhood resulting in long-term isolated anti-HBs OBI carriage as recently described in Chinese blood donors [55]. The frequency of such undetected cases is difficult to evaluate as it largely depends on the sensitivity of the assays used. Recently, data from a large-scale multi-regional study using a comparable HBV screening algorithm showed that 2% and 4% of OBI donors presented no detectable serological markers or an anti-HBs only profile, respectively [56]. The transfusion-transmitted rate of OBI is likely underestimated [57] because HBV DNA might remain undetected despite the use of sensitive NAT assays [19], and due to difficulties in tracing and testing recipients. Our residual risk estimate might be underestimated as well, as it did not take into account seronegative OBI. In England, where anti-HBc testing is not performed on blood donations, a recent study showed that OBI accounted for the majority of HBV residual risk. At least 13 potentially infectious donations from donors with OBI are estimated to remain undetected annually, leading to an overall residual risk of 3.1 per million donations [38]. In France, the HBV screening strategy including anti-HBc and ID-NAT, only anti-HBc and NAT negative eclipse phase donations, potentially contribute to an additional risk that, however, deserves to be estimated.

Interestingly, NAT revealed that 60% (12/20) of HBV DNA only positive donors carried isolated anti-HBs. According to vaccination recording and follow-up investigations, isolated anti-HBs OBI appeared mainly associated to vaccine breakthrough. Similar results were observed in the US in 2011 showing six of the nine NAT only samples that were from donors who had received the HBV vaccine. In five of these six vaccinated donors, a non-A genotype was identified as the dominant strain, whereas sub-genotype A2 (strain found in the HBV vaccine) was the most frequent strain in unvaccinated donors [5]. Furthermore, a recent study in Chinese blood donors reported a prevalence of 9.5% isolated anti-HBs carriers in confirmed OBIs. Chinese isolated anti-HBs OBI were associated with young, vaccinated, adults exposed to HBV genotypes B and C [58]. Similarly, among the nine vaccinated donors from NAT yield cases in our study, seven were infected with a non-A genotype, while the other two were infected with a sub-genotype A2. Remarkably, HBV DNA-only OBIs associated with acute (anti-HBs-) or breakthrough (anti-HBs+) infection were significantly younger than individuals with acute HBV DNA+/HBsAg+/anti-HBc− or anti-HBc+ OBI (3–34 years vs. 45–46 years) for unclear reasons. Overall, these findings may suggest a suboptimal level of protection against infections with non-A2 strains after vaccination, even though HBV vaccination remained protective with a subclinical breakthrough infection [5]. The absence of infectivity of blood in the presence of anti-HBs has been reported in previous studies [59,60,61]. Conversely, blood containing HBV DNA and low-level anti-HBs (<75 mIU/mL) may be at-risk of transmission leading to acute hepatitis [19,22].

## 5. Conclusions

Our experience based on data collected over 21 years of blood donor surveillance provides highlights on HBV infection in blood donors and by extension in the general population in France. HBV infections in blood donors are mostly ethnicity-related chronic hepatitis B. The incidence of infection is very low explaining the current low residual risk which decreased by more than 10-fold compared to the years when no NAT was used. This is consistent with the absence of transfusion transmission declared to the haemovigilance network since the NAT introduction. Our screening strategy based on three HBV markers (HBsAg, anti-HBc, and DNA) prevents the maximum number of infected donations from entering the blood supply but some seronegative OBIs with transiently detectable DNA level might still escape this screening strategy. Such a screening strategy combining three markers remains very expensive so that some studies have questioned abandoning HBsAg [53,62]. In the future, the potential generalization of efficient pathogen-reduction measures to red cell units could lead to an HBV transfusion risk virtually close to zero.

## Figures and Tables

**Figure 1 viruses-14-02507-f001:**
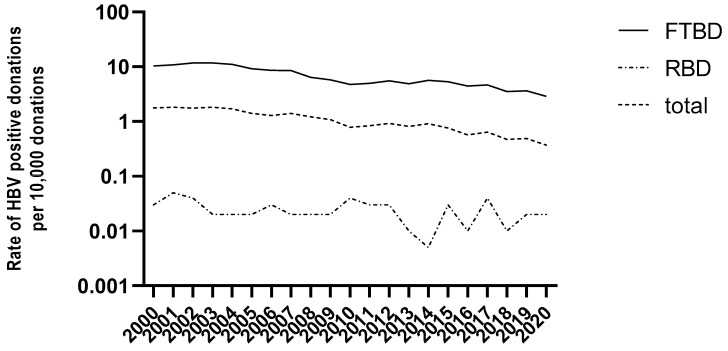
Rates of HBV positive donations (per 10,000) in France between 2000 and 2020 in total blood donor population and by donor status (FTBD: first time blood donors; RBD: repeat blood donors).

**Figure 2 viruses-14-02507-f002:**
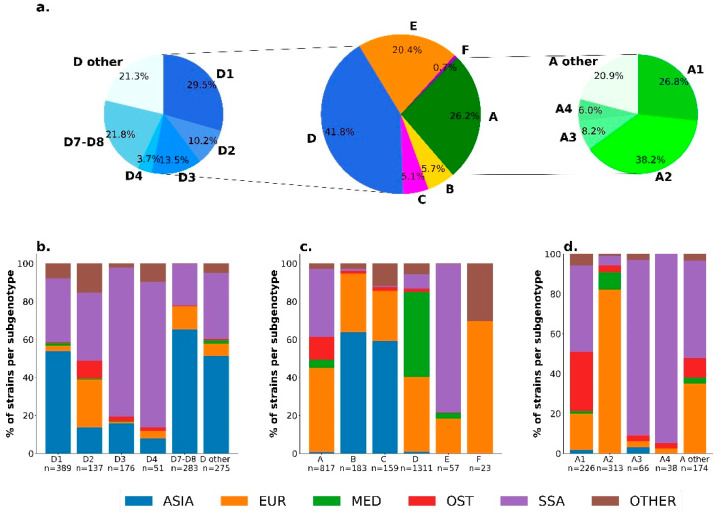
(**a**) Distribution of HBV genotypes, with genotypes A and D sub-genotypes in the total study population. The geographical origin of the donors is specified in (**b**) in the population infected by genotype D (**c**) in the total study population, and (**d**) in the population infected by genotype A.

**Figure 3 viruses-14-02507-f003:**
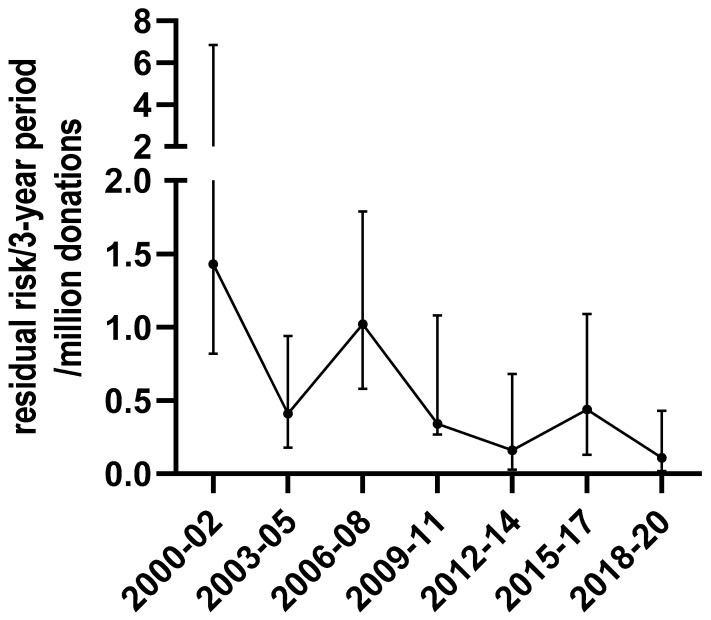
Evolution of the residual risk for HBV, based on the incidence/WP model in blood donations from 2000 to 2020.

**Table 1 viruses-14-02507-t001:** Comparison of demographic and virological data in the four donor populations with positive HBV DNA.

	1. DNA+ /HBsAg+/ Anti-HBc+ (CHB)	2. DNA+ /HBsAg−/ Anti-HBc+ (OBI)	3. DNA+ /HBsAg+/ Anti-Hbc− (Probable ABI)	4. DNA+ /HBsAg−/ Anti-Hbc− (Seronegative OBI)
Total	2079	81	27	25
donor status				
first-time BD	2045 (98%)	69 (85%)	7 (26%)	10 (40%)
repeat BD	18 (1%)	11 (14%)	18 (63%)	15 (60%)
unknown	16 (1%)	1 (1%)	2 (1%)	0 (0%)
*p*-value (2 vs. 3)		<0.0001		
*p*-value (3 vs. 4)			0.5512	
*p*-value (2 vs. 4)		<0.0001		
Age [IC95]				
median	33	54	47	28
minimum	18	20	19	18
maximum	69	69	68	67
*p*-value (2 vs. 3)		0.1739		
*p*-value (3 vs. 4)			0.0616	
*p*-value (2 vs. 4)		<0.0001		
Gender				
males	1597	68	20	14
females	479	13	5	11
unknown	0	0	2	0
sex ratio	3,3	5,2	4,0	1,3
*p*-value (2 vs. 3)		0.7611		
*p*-value (3 vs. 4)			0.1284	
*p*-value (2 vs. 4)		0.006		
Viral load ^1^				
median (log IU/mL)	2.75	0.70	5,24	1.04
minimum (log IU/mL)	0.70	0.70	2.23	0.70
maximum (log IU/mL)	9.53	3.03	8.04	2.65
*p*-value (2 vs. 3)		<0.0001		
*p*-value (3 vs. 4)			<0.0001	
*p*-value (2 vs. 4)		0.0021		
anti-HBs (mIU/mL)				
negative	1510 (4%)^2^	24 (50%)	17 (100%)	8 (40%)
positive	69 (96%)	24 (50%)	0 (0%)	12 (60%)
not tested	497	33	8	5
median	18	55	0	120
Genotype distribution ^3^				
Total genotyped	1593	44	22	20
A	438 (21%)	10 (23%)	13 (55%)	6 (30%)
B	77 (4%)	2 (5%)	2 (9%)	3 (15%)
C	73 (4%)	1 (2%)	1 (5%)	0 (0%)
D	647 (31%)	21 (48%)	2 (9%)	3 (15%)
E	348 (16%)	9 (20%)	4 (18%)	5 (25%)
F	10 (1%)	1 (2%)	1 (5%)	3 (15%)
*p*-value (2 vs. 3)		0.0266		
*p*-value (3 vs. 4)			0.4276	
*p*-value (2 vs. 4)		0.0674		
(1) viral load below the quantification level (6 IU/mL) were considered at 5 IU/mL (0.70 log IU/mL)				
(2) % of the tested samples				
(3) % of genotyped samples				

**Table 2 viruses-14-02507-t002:** Characteristics of the HBV-DNA positive/HBsAg negative/anti-HBc negative infections in blood donors in France from 2005 to 2020.

	Donation								Follow Up		Donor History				Previous Donation	Conclusion (2)
*N*	Year	Donor Status (FTBD/RBD)	Sex	Age	Interdonation Interval (Days)	Viral Load (IU/mL) (1)	Genotype	Anti-HBs (mIU/mL)	HBsAg Seroconversion (Days or Month (M) Post Donation)	Anti-HBc Seroconversion (Days or Month (M) Post Donation)	Anti-HBs (mIU/mL) in Control Samples (Days Post Donation)	Vaccination (Year)	Origin	Risk Factor		
1	2007	FTBD	M	28	-	8	A	NT	Yes (Day 20)	No (day 20)	NT	?	Caribbean	Not investigated	-	Acute
2	2010	RBD	M	28	136	44	E	414	NT	NT	NT	?	France	Not investigated	Anti-HBs negative in 2009	Probable vaccine breakthrough infection
3	2010	FTBD	M	24	-	NT	NT	NT	NT	NT	NT	?	Not investigated	Not investigated	-	?
4	2011	RBD	M	36	112	5	A2	Negative	No (M5)	Yes (IgM) (M5)	Yes (M4)	incomplete (2004)	France	MSM	HBV-DNA negative	Vaccine breakthrough infection
5	2011	FTBD	F	19	-	38	E	Negative	Yes (Day 25)	No (Day 25)	NT	?	Andorra	Partner HBV	-	Acute
6	2012	RBD	M	32	238	60	D1	380	NT	NT	NT	Probable (health care professional)	France	Partner from endemic area	HBV-DNA negative	Probable vaccine breakthrough infection
7	2012	RBD	F	19	204	456	E	Negative	Yes (Day 5)	NT	NT	?	France	Partner from endemic area		Acute
8	2012	RBD	F	57	56	NT	E	13	No (M4, M6)	Yes (M6)	6800 (M6)	?	France	Partner from endemic area	Anti-HBs negative in 2011	Acute
9	2012	FTBD	F	18	-	NT	NT	NT	No (Day 17)	No (Day 17)	NT	?	France	Piercing	-	?
10	2014	FTBD	F	18	-	NT	A1	Negative	No (Day 14)	No (Day 14)	8000 (M10)	Yes (2004)	France	Partner from endemic area	-	Vaccine breakthrough infection
11	2014	FTBD	M	18	-	14	B	16	No (Days 22, 42)	No (Days 22, 42)	185 (Day 22) 188m (Day 42)	Yes (1997)	France	Partner from endemic area	-	Probable vaccine breakthrough infection
12	2014	RBD	F	43	1393	NT	NT	NT	No (Day 15)	No (Day 15)	NT	?	Not investigated	Not investigated	Not investigated	?
13	2015	RBD	F	25	314	5	E	216	No (Day 11)	Yes (IgM) (Day 11)	>25000 (Day 11)	Yes (1995)	France	Unknown	Not investigated	Vaccine breakthrough infection
14	2015	RBD	M	34	614	108	A2	Negative	Yes (Day 18)	No (Day 18)	NT	?	France	Unknown	Not investigated	Acute
15	2015	RBD	M	20	395	5	D1	193	No (Day 28)	No (Day 28)		Yes (1995)	France	Unknown	Not investigated	Vaccine breakthrough infection
16	2015	RBD	M	49	104	8	A2	20	No (M 3, M4)	Yes (M 3, M4)	376 (M4)	Yes (1999)	France	Unknown	HBV-DNA negative	Probable vaccine breakthrough infection
17	2015	FTBD	M	20	-	25	B4	14	No (M 8)	No (M 8)	NT	Yes (date unknown)	?	Unknown	-	Vaccine breakthrough infection
18	2016	FTBD	M	41	-	5	D3	Negative	No (Day 9)	No (Day 9)	NT	?	Iran	Sexual	-	?
19	2016	RBD	F	46	453	18	F1	Negative	Yes (M 3)	No (M 3)	NT	No	France	Sexual	Not investigated	Acute
20	2016	FTBD	F	19	-	NT		255	No (Day 13)	Yes (Day 13)	1177 (Day 13)	?	France	partner from endemic area	-	Probable vaccine breakthrough infection
21	2017	RBD	M	65	153	NT	F1	NT	No (Day 25)	No (Day 25)	8 (Day 7) 2824 (Day 25)	?		Unknown	HBV-DNA negative	Probable vaccine breakthrough infection
22	2019	RBD	M	68	182	5	A2	275	No (Day 39)	No (Day 39)	606 (Day 18)	No	France	Unknown	HBV-DNA positive (see OBI case reports)	OBI
23	2020	FTBD	F	20	-	5	B4	47	No (Day 13)	No (Day 13)	52 (Day 13)	Yes (chidlhood)	France	Residence in China in early years of life	-	OBI
24	2020	RBD	F	58	112	5	Not typable	Negative	No (Day 67)	Yes (IgM) (Day 67)	NT	No		Partner HBV	HBV-DNA negative	Acute
25	2020	FTBD	M	24	-	234	F1	11	No (Day 5)	No (Day 5)	NT	incomplete (7 months before donation)	France	sexual	-	Probable vaccine breakthrough infection

FTBD: first time blood donor, RBD: repeat blood donor, F: female, M: male, NT: not tested. (1) When Viral Load <6 IU/mL noted 5 IU/mL. (2) Acute infection was defined by HBsAg and/or anti-HBc (with or without anti-HBc IgM antibodies) and/or anti-HBs (in the absence of vaccination) in the subsequent sample. If seroconversion was not observed in the follow up sample or if previous donation was HBV DNA positive in the archived sample, the donation was classified as OBI. A donation positive for anti-HBs was classified as vaccine breakthrough infection if the donor was previously vaccinated (even partially) and as probable breakthrough infection if the vaccination status was unknown but the donation was positive for anti-HBs.

## Data Availability

Not applicable.

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
