# Peer review of "Insights on 21 Years of HBV Surveillance in Blood Donors in France"

_viruses, 2022, doi:10.3390/v14112507_

Round 1
Reviewer 1 Report
Dear authors,
Please consider the following minor issues:
1)Abstract, line 22, "HBV positive": Do you mean "HBsAg positive"? Or do you mean HBsAg positive and/or HBV-DNA positive? I propose to apply a strict textual regimen to the entire paper, for example: "HBV positive" means including all HBsAg positive and/or HBV-DNA positive samples.
2)Abstract, line 25, "was divided by 13": consider the more common and more correct "decreased 13-fold". (It was not really divided by someone).
3)Introduction, line 83/84, "without compromising blood availability": Are you sure that in high-prevalent countries anti-core screening indeed would cause too much donor loss, if core-positive donors with anti-HBs >100 U/L are allowed to donate? We must realise that especially in high prevalent countries core-Ab screening would have a significant yield. Maybe this higher yield would make a limited donor loss (only core-pos donors with anti-s <100) acceptable?
4)Methods, line 170 "Chronic infection was defined": I think this definition erroneously will include some acute infections that are resolving: HBV-DNA is about to disappear, while anti-core already is becoming detectable?
5) Results, line 211, "HBV reactive" = "HBsAg and/or HBV-DNA reactive" ?
6) Results, line 212: "been divided by 4.8" -> "decreased 4.8-fold".
7) Results, line 213: "by 3.5" -> "3.5-fold"
8) Results, line 217/218: "was divided by a factor of 2.8" -> "decreased 2.8-fold".
9) Results, line 221-222, Figure 1: Consider to redesign the graph, applying one logarithmic Y-axis, ranging from 0.01 to 20. Maybe this makes it possible to combine the 3 'fragmented' Y-axes into one Y-axis?
10) Results, line 372, "was divided by 13" -> "decreased 13-fold".
11) Discussion, line 379/380, "tested HBV positive for HBV in France" -> "tested positive for HBsAg and/or HBV-DNA in France" ?
12 Discussion: What was the overall yield of HBsAg testing, for the donations that were also tested for HBV-DNA and anti-core? Could HBsAg testing be abolished when anti-core and HBV-DNA testing is applied?
With kind regards.
Author Response
REVIEWER I
1) Abstract, line 22, "HBV positive": Do you mean "HBsAg positive"? Or do you mean HBsAg positive and/or HBV-DNA positive? I propose to apply a strict textual regimen to the entire paper, for example: "HBV positive" means including all HBsAg positive and/or HBV-DNA positive samples.
=> this is indeed what is meant here, throughout the manuscript when “HBV positive” is used. As noted in the methods section, HBV NAT was not performed from 2000 to 2005.
=> we added the definition (HBsAg+ and/or HBV DNA+ in the introduction (line 119) and in the results section (line 213)
2) Abstract, line 25, "was divided by 13": consider the more common and more correct "decreased 13-fold". (It was not really divided by someone).
=> corrected
3) Introduction, line 83/84, "without compromising blood availability": Are you sure that in high-prevalent countries anti-core screening indeed would cause too much donor loss, if core-positive donors with anti-HBs >100 U/L are allowed to donate? We must realise that especially in high prevalent countries core-Ab screening would have a significant yield. Maybe this higher yield would make a limited donor loss (only core-pos donors with anti-s <100) acceptable?
- In low prevalence countries, anti HBc positive donors are allowed to donate if they also have a sufficient level of anti HBs (>500 UI/ml in France) but only for plasma for fractionation.
4) Methods, line 170 "Chronic infection was defined": I think this definition erroneously will include some acute infections that are resolving: HBV-DNA is about to disappear, while anti-core already is becoming detectable?
=> Indeed. A chronic infection is defined by the presence of HBs Ag or DNA for more than 6 months. We do not have the follow-up of every positive donor, so we cannot really validate this fact. Nevertheless, 85 % of our samples are anti-HBe positive, even if not shown in the paper. They are probably chronic infection in the vast majority if cases.
Even if “resolving infections”, their detection will depend on the sensitivity of HBV DNA screening in blood.
5) Results, line 211, "HBV reactive" = "HBsAg and/or HBV-DNA reactive" ?
=> "HBsAg and/or HBV-DNA reactive" (precision added)
6) Results, line 212: "been divided by 4.8" -> "decreased 4.8-fold".
=> corrected
7) Results, line 213: "by 3.5" -> "3.5-fold"
=> corrected
8) Results, line 217/218: "was divided by a factor of 2.8" -> "decreased 2.8-fold".
=> corrected
9) Results, line 221-222, Figure 1: Consider to redesign the graph, applying one logarithmic Y-axis, ranging from 0.01 to 20. Maybe this makes it possible to combine the 3 'fragmented' Y-axes into one Y-axis?
=> done.
10) Results, line 372, "was divided by 13" -> "decreased 13-fold".
=> corrected
11) Discussion, line 379/380, "tested HBV positive for HBV in France" -> "tested positive for HBsAg and/or HBV-DNA in France"?
=> precision added
12 Discussion: What was the overall yield of HBsAg testing, for the donations that were also tested for HBV-DNA and anti-core? Could HBsAg testing be abolished when anti-core and HBV-DNA testing is applied?
=>This is a very interesting point. HBsAg +/ anti HBc -/ DNA - donations were not included in our study. In our experience, the majority of such donations were from donors with a history of HBV vaccination shortly before their donation but information was not always available and we assume that the remaining donations were falsely positive for HBsAg even though we cannot prove this hypothesis. Our data are not sufficient to answer this question.
In a study performed in blood donors in the US (Dodd et al Transfusion 2018;58;2166–2170) it was shown that the vast majority of HBsAg-only donations could be considered as false positive. But some of them were DNA pos at least once when tested in several replicates. The overall frequency of this type of sample was one in 4.4 million donations in the US at that time. But the authors underlined that it is certainly unclear whether such donations would be infectious, with such a low level of HBV DNA.
Reviewer 2 Report
In the Introduction: Remind of HBV vaccine campaigns in France (dates ~1980s-1990s) to later explain young age of donors with OBI Line 46 - classified Line 145 - mL vs. ml (line 270, 283, 438-483 Line 189 - add specific Reference for window period of 22 days.
Line 189 - list the 7 periods.
Line 190-192: clarify which 3-year period and the criteria for donor exclusion Line 220-221: PY (per year?) Figure 1: Y axis - check the unit which is labelled as "per 10,000" but reporting FTD per 100,000
• FTD: 8.13/100,000
• RD: 0. 02/10,000
Figure 2: increase font
Line 167: Introduce the definitions acronyms used in results (CHB etc...)
Table 1: change headers, remove HBV and make it an overall "HBV infection markers" and change headers to CHB, OBI, Acute, sero-negative OBI Figure 3: Correct period 2012-2014 (now showing 2002- 2014) and change Y scale and expand 0-1 data Line 410: remove the in : the risk of "the" viral transmission by transfusion Line 461-463: incomplete sentence.
Author Response
REVIEWER II
In the Introduction: Remind of HBV vaccine campaigns in France (dates ~1980s-1990s) to later explain young age of donors with OBI
=> HBV vaccination was not mandatory, therefore, we prefered not mentioning this fact.
Line 46 - classified
=> corrected
Line 145 - mL vs. ml (line 270, 283, 438-483)
=> corrected to “mL” thoughout the text
Line 189 - add specific Reference for window period of 22 days.
=> added
Line 189 - list the 7 periods.
=> modified to “seven periods of three years”
Line 190-192: clarify which 3-year period and the criteria for donor exclusion
=> these lines now read: “ Repeat donors with a donation collected in a previous 3-year period who were found HBV-DNA positive in the lookback investigation were excluded from the calculation of the risk, as they were therefore not incident cases” .
Line 220-221: PY (per year?)
=> precision added: person-year
Figure 1: Y axis - check the unit which is labelled as "per 10,000" but reporting FTD per 100,000
- FTD: 8.13/100,000
- RD: 0. 02/10,000
=> the prevalence is given per 10,000 BD (for total BD population and RBD and FTBD). This incidence, calculated in the RBD in 3-year person is given per 100,000 person-year and is not shown on the graph, but in the supplementary Table 1.
Figure 2: increase font
=> done
Line 167: Introduce the definitions acronyms used in results (CHB etc...)
=> done
Table 1: change headers, remove HBV and make it an overall "HBV infection markers" and change headers to CHB, OBI, Acute, sero-negative OBI=> Table 1 header must read “HBV-DNA+ / HBsAG +/anti-HBc+” etc, and not HBV.
=> we removed “HBV” before “DNA”, to avoid the confusion and added the definitions in brackets.
Figure 3: Correct period 2012-2014 (now showing 2002- 2014) and change Y scale and expand 0-1 data
=> done
Line 410: remove the in : the risk of "the" viral transmission by transfusion
=> corrected
Line 461-463: incomplete sentence.
=> sentence modified